# Prospective validation of smartphone-based heart rate and respiratory rate measurement algorithms

Sean Bae[1,8], Silviu Borac[1,8], Yunus Emre[1,8], Jonathan Wang[1,8], Jiang Wu[1,8], Mehr Kashyap[2], Si-Hyuck Kang [1,3✉], Liwen Chen[1], Melissa Moran[1], Julie Cannon[2], Eric S. Teasley[1], Allen Chai[1], Yun Liu [1✉], Neal Wadhwa[4], Michael Krainin[4], Michael Rubinstein[4], Alejandra Maciel[1], Michael V. McConnell [1,5], Shwetak Patel[1,6], Greg S. Corrado[1], James A. Taylor[1,6,9], Jiening Zhan[1,9✉] & Ming Jack Po[1,7,9]

## Abstract

**Background** Measuring vital signs plays a key role in both patient care and wellness, but can be challenging outside of medical settings due to the lack of specialized equipment.

**Methods** In this study, we prospectively evaluated smartphone camera-based techniques for measuring heart rate (HR) and respiratory rate (RR) for consumer wellness use. HR was measured by placing the finger over the rear-facing camera, while RR was measured via a video of the participants sitting still in front of the front-facing camera.

**Results** In the HR study of 95 participants (with a protocol that included both measurements at rest and post exercise), the mean absolute percent error (MAPE) ± standard deviation of the measurement was 1.6% ± 4.3%, which was significantly lower than the pre-specified goal of 5%. No significant differences in the MAPE were present across colorimeter-measured skin-tone subgroups: 1.8% ± 4.5% for very light to intermediate, 1.3% ± 3.3% for tan and brown, and 1.8% ± 4.9% for dark. In the RR study of 50 participants, the mean absolute error (MAE) was 0.78 ± 0.61 breaths/min, which was significantly lower than the pre-specified goal of 3 breaths/min. The MAE was low in both healthy participants (0.70 ± 0.67 breaths/min), and participants with chronic respiratory conditions (0.80 ± 0.60 breaths/min).

**Conclusions** These results validate the accuracy of our smartphone camera-based techniques to measure HR and RR across a range of pre-defined subgroups.

## Plain language summary

Accurate measurement of the number of times a heart beats per minute (heart rate, HR) and the number of breaths taken per minute (respiratory rate, RR) is usually undertaken using specialized equipment or training. We evaluated whether smartphone cameras could be used to measure HR and RR. We tested the accuracy of two computational approaches that determined HR and RR from the videos obtained using a smartphone. Changes in blood flow through the finger were used to determine HR; similar results were seen for people with different skin tones. Chest movements were used to determine RR; similar results were seen between people with and without chronic lung conditions. This study demonstrates that smartphones can be used to measure HR and RR accurately.

[1] Google Health, Palo Alto, CA, USA. [2] Google Health via Advanced Clinical, Deerfield, IL, USA. [3] Seoul National University Bundang Hospital, Seongnam-si, South Korea. [4] Google Research, Cambridge, MA, USA. [5] Stanford University, Stanford, CA, USA. [6] University of Washington, Seattle, WA, USA. [7] Present address: Ansible Health, Mountain View, CA, USA. [8] These authors contributed equally: Sean Bae, Silviu Borac, Yunus Emre, Jonathan Wang, Jiang Wu. [9] These authors jointly supervised this work: James A. Taylor, Jiening Zhan, Ming Jack Po. ✉email: eandp303@snu.ac.kr; liuyun@google.com; jieningz@gmail.com

Measurement of heart rate (HR) and respiratory rate (RR), two of the four cardinal vital signs—HR, RR, body temperature, and blood pressure—is the starting point of the physical assessment for both health and wellness. However, taking these standard measurements via a physical examination becomes challenging in telehealth, remote care, and consumer wellness settings[1–3]. In particular, the recent COVID-19 pandemic has accelerated trends towards telehealth and remote triage, diagnosis, and monitoring[4,5]. Although specialized devices are commercially available for consumers and have the potential to motivate healthy behaviors[6], their cost and relatively low adoption limit general usage.

On the other hand, with smartphone penetration exceeding 40% globally and 80% in the United States[7], up to 3.8 billion individuals already have access to a myriad of sensors and hardware (video cameras with flash, accelerometers, gyroscope, etc.) that are changing the way people interact with each other and their environments. A combination of these same sensors together with novel computer algorithms can be used to measure vital signs via consumer-grade smartphones[8–12]. Indeed, several such mobile applications ("apps") are available, some with hundreds of thousands of installs[13]. However, these apps seldom undergo rigorous clinical validation for accuracy and generalizability to important populations and patient subgroups.

In this work, we present and validate two algorithms that make use of smartphone cameras for vital sign measurements. The first algorithm leverages photoplethysmography (PPG) acquired using smartphone cameras for HR measurement. PPG signals are recorded by placing a finger over the camera lens, and the color changes captured in the video are used to determine the oscillation of blood volume after each heart beat[14]. In the second algorithm, we leverage upper-torso videos obtained via the front-facing smartphone camera to track the physical motion of breathing to measure RR. Herein, we describe both the details of the algorithms themselves and report on the performance of these two algorithms in prospective clinical validation studies. For the HR study, we sought to demonstrate reliable and consistent accuracy on diverse populations (in terms of objectively measured skin tones, ranging from very light to dark skin), whereas for the RR study, we aimed to demonstrate robust performance in subgroups with and without chronic respiratory conditions. This study confirms our smartphone camera-based techniques are accurate in measuring HR and RR across a range of predefined subgroups.

## Methods

We conducted two separate prospective studies (Table 1) to validate the performance of two smartphone-based algorithms, one for measurement HR and the other for RR measurement (Fig. 1). The user interfaces of the two custom research apps are shown in Supplementary Fig. 1. The HR algorithm measured PPG signals via videos of the finger placed over the rear camera, while the RR algorithm measured movements of the chest via videos captured from the front camera. Next, we provide more details on each algorithm and corresponding study.

### HR measurement

*Algorithm description.* Prior work in computer vision to extract heart rate from RGB (red-green-blue) video signals has leveraged manually extracted features in PPG signals from the finger for arrhythmia detection[15], ballistocardiographic movements from fingertips[16], red-channel PPG from fingertip videos[17], and the relationship between RGB channels[18].

Our method estimates HR by optically measuring the PPG waveform from participants' fingertips and then extracting the

**Table 1 Baseline characteristics of the study participants.**

|  | Heart rate (HR) study | Respiratory rate (RR) study |
|---|---|---|
| No. participants analyzed | 95 | 50 |
| No. recordings | 352 | 50 (for each algorithm version*) |
| Age (mean ± standard deviation) | 41.8 ± 15.0 | 50.0 ± 16.0 |
| Age groups |  |  |
| <40 years | 41 (43%) | 17 (34%) |
| 40–59 years | 39 (41%) | 21 (42%) |
| ≥60 years | 15 (16%) | 12 (24%) |
| No. female (%) | 71 (75%) | 26 (52%) |
| No. male (%) | 24 (25%) | 24 (48%) |
| Race/ethnicity: *n*, % |  |  |
| White, non-Hispanic | 25 (26%) | 18 (36%) |
| White, Hispanic | 0 (0%) | 22 (44%) |
| Black, non-Hispanic | 61 (64%) | 6 (12%) |
| Black, Hispanic | 0 (0%) | 1 (2%) |
| Asian/pacific islander | 7 (7%) | 3 (6%) |
| Multiple races, non-Hispanic | 1 (1%) | 0 (0%) |
| Multiple races, Hispanic | 1 (1%) | 0 (0%) |
| Measured skin tone**: *n* (%) |  |  |
| 1 (Fitzpatrick types 1–3) | 31 (33%) | N/A |
| 2 (Fitzpatrick types 4–5) | 32 (34%) |  |
| 3 (Fitzpatrick type 6) | 32 (34%) |  |
| Chronic respiratory conditions: *n* (%) |  |  |
| None | N/A | 10 (20%) |
| Asthma |  | 33 (66%) |
| COPD |  | 4 (8%) |
| Both |  | 3 (6%) |

*COPD* chronic obstructive pulmonary disease.
*RR was measured twice, once for each one of two algorithm versions (see Methods).
**Measurements were done on the cheek using a Pantone RM200QC Spectro (see Methods).

dominant frequency. First, several rectangular regions of interest (ROI) were manually selected from the video frames (linear RGB at 15 frames per second and at a resolution of $640 \times 480$ pixels). The chosen ROIs were the full-frame, the left half, the right half, the top half, and the bottom half of the frames. Since camera pixels are illuminated non-homogeneously, signal strength can have spatial variations across pixels[19]. Our method simultaneously analyzes different ROIs to identify one with the greatest SNR.

Pixels in each ROI were averaged per channel to reduce the effects of sensor and quantization noise, similar to prior work[18]. The pulsatile blood volume changes were present as the AC components in these smoothed signals. We then weighted the three RGB waveforms to predict a single PPG waveform (after an empirical grid search across all 3 channels: RGB, we arrived at weights 0.67, 0.33, and 0, respectively) for each ROI.

The resulting PPG waveforms were bandpass filtered to remove low- and high-frequency noise unlikely to be valid HR. Filter cut-off frequencies corresponded to a low of 30 beats/min and a high of 360 beats/min. Next, large amplitude changes in PPGs due to motion were suppressed by limiting maximum allowed changes in amplitudes to 3 times of the moving average value. Then, frequency-domain representations of PPGs were computed using the Fast Fourier Transform (FFT), from which we identified the dominant frequencies with maximum power. Because the PPG signals are periodic with multiple harmonics, the powers of the base frequencies were computed by summing the powers of their first, second, and third harmonics. SNRs were estimated for each ROI by computing the ratio between the power of the dominant

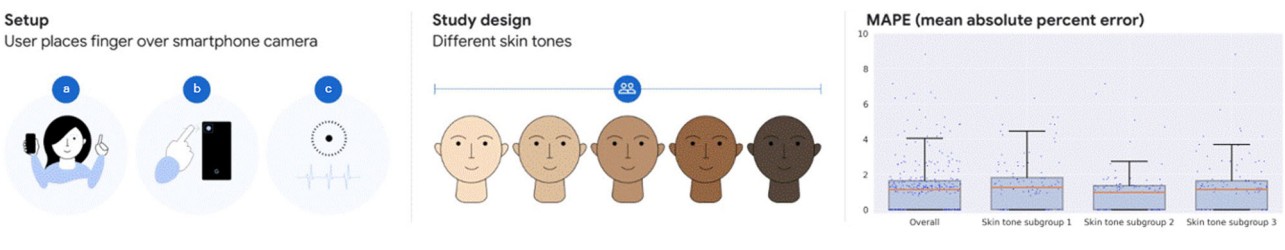

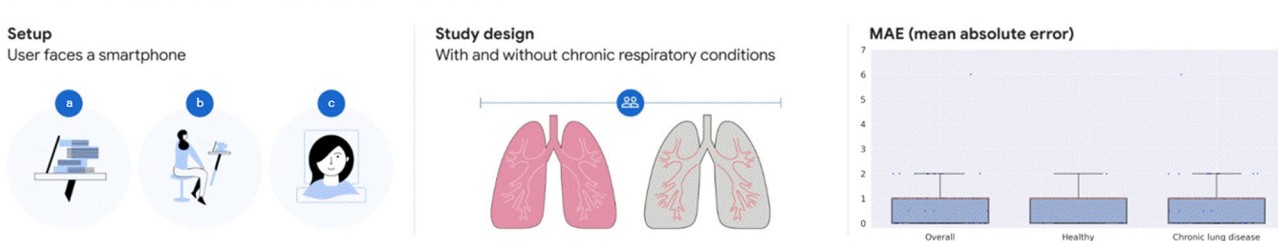

**Fig. 1 Smartphone-based monitoring of two key vital signs: heart rate (HR) and respiratory rate (RR).** Setup of how measurements are taken: with the finger over the rear-facing camera for HR (upper panel) and using a video of the participant via the front-facing camera for RR (lower panel). Study design: to ensure generalization across skin tones for HR (n = 95) and generalization to participants with chronic respiratory conditions (chronic obstructive pulmonary disease and asthma) for RR (n = 50). Skin-tone subgroup 1 (n = 31) corresponds to Fitzpatrick skin types 1–3 (very light, light, and intermediate); subgroup 2 (n = 32) corresponds to types 4–5 (tan and brown); and subgroup 3 (n = 32) corresponds to type 6 (dark). RR study included healthy participants (n = 10) and participants with chronic respiratory conditions (n = 40). Metrics: the main measurements were mean absolute percent error (MAPE) for HR and mean absolute error (MAE) for RR. In the boxplots, the orange lines and box edges indicate the quartiles; whiskers indicate 1.5 times the interquartile range beyond the upper and lower quartiles; dots indicate individual data points (average percent error or absolute error). For the HR study, five outlier data points in the "overall" group extend beyond the axes (>10%) and are not shown; these outliers are distributed across the three skin tone subgroups (1, 2, and 2, respectively). All data points are shown for the RR study.

frequency and the powers of non-dominant frequencies on a logarithmic scale. ROIs were filtered to only those with a SNR ≥0 dB, and the dominant frequency of the ROI with the highest SNR was reported. If no such ROI existed, no HR was reported. Further details are provided in Supplementary Methods.

*Study design and participants.* We performed a prospective observational clinical validation study to assess the accuracy of the study algorithm in estimating HR in individuals of diverse skin tones (Supplementary Fig. 1a). Participants were enrolled at a clinical research site (Meridian, Savannah, GA) from October 2020 to December 2020. Study eligibility criteria were limited to excluding participants with significant tremors or inability to perform physical activity. The inclusion/exclusion criteria are detailed in Supplementary Table 1a. Study enrollment was stratified into three skin-tone subgroups (mapped to Fitzpatrick skin types[20]; see Supplementary Table 2) to ensure broad representation: (1) types 1–3 (very light, light, and intermediate); (2) types 4–5 (tan and brown), and (3) type 6 (dark). Skin tone was objectively measured from the participants' cheek skin using an RM200QC Spectro colorimeter (X-Rite, Grand Rapids, MI). Evidence suggests that darker skin tone is frequently underrepresented in medical datasets[21], and that medical devices using optical sensors may be less accurate in those individuals[22–24]. Therefore, the darkest skin-tone subgroup was intentionally oversampled to ensure the algorithm's unbiased performance over various skin tones. Informed consent was obtained from all study participants in accordance with the tenets of the Declaration of Helsinki. The study protocol was approved by Advarra IRB (Columbia, MD; protocol no. Pro00046845). The clinical research site followed standard safety precautions for COVID-19 in accordance with the Centers for Disease Control and Prevention guidelines.

*Data collection.* Each participant underwent four 30-s data collection episodes with their index finger (of a hand of their choice) held directly over the study phone camera. Three of the 30-s episodes were collected at rest under various ambient brightness/ lighting conditions: (1) with camera flash on and under normal ambient light, (2) with the flash off and under normal ambient light, and (3) with the flash off and under dim light. The fourth episode was collected post-exercise. In the original protocol, participants were instructed to ride a stationary bicycle for 30 s as strenuously as possible against light to medium resistance. After enrolling 37 participants, the exercise protocol was modified (with an IRB amendment) to achieve higher participant HR: participants were encouraged to achieve 75% of their maximal HR, which was calculated by subtracting the participant's age from 220 beats/min. The exercise was completed either when the goal HR was achieved or when the participant asked to stop. The data were collected with the flash off and under normal ambient light. Lighting conditions were controlled using two overhead and one front light-emitting diode (LED) lights. The brightness level of the study environment was measured by a Lux meter (LT300 Light Meter, Extech, Nashua, NH) prior to each study. Measured brightness values were between 160 and 200 Lux for normal ambient light, and between 95 and 110 Lux for dim light.

The study was conducted using a mobile app deployed to a Pixel 3 smartphone running Android 10 (Google LLC, Mountain View, CA). HR estimation using the app was generally completed by the study participants following the in-app instructions, with the coordinators providing feedback on usage when needed. The reference HR was measured simultaneously during each data collection episode using a Masimo MightySat® (Masimo, Irvine, CA), which is US Food and Drug Administration-cleared for fingertip measurement of pulse rate[25]. The measurements were conducted in accordance

with the manufacturer's manual and taken at the end of each episode.

*Statistics and reproducibility.* Each participant contributed up to three HR measurements at rest (with different lighting conditions), and up to one post-exercise. Measurements were paired observations: the algorithm-estimated HR and the reference HR from the pulse oximeter. For each algorithm measurement, up to three tries were allowed, and the number of tries required was recorded. The baseline characteristics of the participants in whom a valid measurement for HR could not be obtained were compared to those of participants for whom a valid measurement was obtained using Fisher's exact test. A paired measurement was dropped if either the algorithm estimation or reference measurement failed. The absolute error of each paired measurement was calculated as the absolute value of the difference between the algorithm-estimated and reference HR values. The MAE was the mean value of all absolute errors. Similarly, the absolute error from each paired measurement was divided by the reference value for that measurement and multiplied by 100 to produce the absolute percentage error. The MAPE was the mean value for all absolute percent error values. The standard deviation of MAPE was calculated; no adjustment for multiple observations was made since the effects of clustering were negligible.

The MAPE was the primary study outcome, as recommended by the current standards for HR monitoring devices[26]. We also computed the standard deviation and 95th percentiles. Sign tests were used to determine whether the absolute percentage errors were significantly <5%, both for the entire group of participants and the three skin-tone subgroups; data from individual data windows were analyzed separately. Bland–Altman plots were used to visualize the agreement between the estimated values and the reference measurements and assess for any proportional bias (trends in the error with increasing values)[27]. The mean differences were derived from the random-effects model considering the repeated measurement nature of the samples. For samples that did not follow a normal distribution based on a Shapiro-Wilk test, the 2.5th and 97.5th percentiles were provided as the limits of agreement. The subgroup analysis across the three skin-tone subgroups was pre-specified.

*Sample size calculation.* HR data collection was planned for ~100 participants. Enrollment up to a maximum of 150 participants was allowed as we anticipated that some enrolled participants would be excluded prior to contributing HR data because they failed to meet the required skin tone distribution or because they were not able to exercise. Requirements for participant enrollment termination included ≥60 paired HR measurements in the dark skin tone subgroup and ≥20% of the post-exercise reference HR >100 beats/min. The study hypothesis was that MAPE was <5% in all of the three skin-tone subgroups. To estimate the sample size required for the study, we first conducted an IRB-approved feasibility study with a different set of 55 participants and similar measurements both at rest and post-exercise. In that study, the MAPE ± standard deviation was 0.91 ± 3.68%. Assuming double the mean and SD (i.e., 1.82 and 7.36%, respectively), a minimum of two paired measurements per participant, a skin-tone subgroup of ~25 participants, and some dropout from incomplete data, the power to detect a MAPE >5% was >0.8.

## RR measurement
*Algorithm description.* Prior work in computer vision and sensors to extract RR from RGB video signals relied on changes in color intensities at specific anatomical points[28,29], tracking head

motions[30,31], estimating optical flow along image gradients[32], or factorizing the vertical motion matrix[33].

Our contactless method estimates RR by performing motion analysis in an ROI of the video stream and requires that the face and upper torso be in the video frame. A previously described face detector[34] is used to obtain a set of face landmarks defining the contour of the face, and the bounding box for the face is computed from the face contour. Subsequently, an ROI around the upper torso is computed by extrapolating from the bounding box of the face. A simple extrapolation method that uses just constant coefficients was shown to be robust to variations in head and torso size. The height and width of the torso ROI is set to 1.4 and 2.5 times the face ROI height and width, respectively. At this point, the upper torso ROI is an RGB image. To attain a frame rate of 15 frames per second this RGB image is converted to a luma-only image and resampled to a size of 15k pixels while maintaining the same aspect ratio.

The main challenge was that variations in the video due to respiratory motions are hard to distinguish from noise. We build on Eulerian, phase-based motion processing[35] that is particularly suited for analyzing subtle motions. In each video frame, the position at each pixel was represented by the phase of spatially localized sinusoids in multiple scales (frequencies). To aggregate the information across scales and to obtain an intuitive representation of motion, we then transformed the spatial phases into optical flow by linearly approximating the position implied by each phase coefficient and averaging across scales. Using the Halide high-performance image library[36], we were able to speed up the phase and optical flow computation to achieve real-time processing (1–4 ms per frame on Pixel 3a and Pixel 4 mobile devices).

It turns out that for estimating the respiratory rate it is sufficient to analyze only the vertical component of the optical flow, so only the vertical component was processed in the subsequent steps. Ensembling was then used to improve the predictive performance. A spectral-spatial ensemble was built in the following way. The respiratory ROI, together with the four quadrants obtained by equally subdividing the ROI defined five regions over which the vertical component of the optical flow was averaged. This resulted in five respiratory waveforms. Next, frequency-domain representations for each of these respiratory waveforms were computed via FFT, from which power spectra were computed. The number of samples in the rolling FFT transform is 900 which provides sufficient resolution for the respiratory rate in the [6, 60] breaths/min range at a video rate of 15 frames per second. The power spectra corresponding to the five regions were then aggregated (added) to obtain a final ensembled power spectrum. Bandpass-filtering was performed to remove low and high frequencies unlikely to represent valid RRs. Filter cut-off frequencies corresponded to a low of 6 breaths/min and a high of 60 breaths/min. The maximum power frequency and the corresponding SNR value were computed from the ensembled power spectrum. The waveform corresponding to the entire ROI is used for displaying the breathing pattern to the user in the mobile app.

Often there was insufficient periodicity in the respiratory waveform (e.g., the participant briefly held their breath or changed their respiratory rate within the time window used for analyzing the waveform). To increase the robustness of RR estimation, the algorithm falls back on a time-domain estimation method based on counting zero crossings of the waveform corresponding to the entire ROI whenever the SNR obtained via the FFT-based method was lower than a certain threshold. We tested two versions of the algorithm, differing only in terms of this threshold: SNR $< -6.0$ dB (version A) and SNR $< -4.0$ dB (version B). The higher value for the threshold in version B

invoked the time-domain estimation method more often, which was hypothesized to improve accuracy by improving robustness to irregular breathing.

*Study design and participants.* We performed a prospective observational clinical validation study to assess the accuracy of the study algorithm in measuring the RR in healthy adults and patients with chronic respiratory conditions (Supplementary Fig. 1b). Participants were enrolled at a clinical research site (Artemis, San Diego, CA) between June 2020 and July 2020. Chronic respiratory conditions included moderate or severe COPD and asthma that was not well-controlled based on specific study criteria (Supplementary Table 1b). Also, participants with significant tremors were excluded. Further details and criteria are presented in Supplementary Table 1b. Informed consent was obtained from all study participants in accordance with the tenets of the Declaration of Helsinki. The study protocol was approved by Aspire IRB (now WCG IRB, Puyallup, WA; protocol no. 20201594). The clinical research site followed standard safety precautions for COVID-19 in accordance with the Centers for Disease Control and Prevention guidelines.

*Data collection.* Each participant underwent 30 s of data collection using a Pixel 4 smartphone running Android 10 (Google LLC, Mountain View, CA). The two algorithm versions (A and B) were tested sequentially. The participants followed the study protocol via instructions from the study app, without intervention from the study staff. Participants were prompted to prop the study phone on a table using provided common household items, such that the upper body was centered in the video capture (Fig. 1). There were no specific requirements on the type of clothing worn during the study or additional custom lighting equipment. The in-app instructions guided the participants to wait several minutes after any active movement. The participants were encouraged to stay comfortable and breathe normally during 30 s of measurement.

During the data collection, RR was manually counted and recorded by two research coordinators. The two observers counted the number of breaths independently and were blinded to the algorithm-estimated results. The agreement between the two measurements was high (Pearson correlation coefficient: 0.962; mean difference: 0.48 ± 0.88 breaths/min; range, 0–4). The mean of the two human-measured RRs rounded off to the nearest integer, was taken to be the reference RR.

*Statistics and reproducibility.* Each participant contributed a single pair of measurements for each algorithm version, and the MAE was used as the primary evaluation metric. The study hypothesis was that MAE would be <3 breaths/min. One-sample t-tests were done to determine whether the MAE was statistically significantly <3 breaths/min. A prespecified subgroup analysis was also performed, stratified by history of chronic respiratory conditions. In addition, post hoc subgroup analyses were performed for age and race/ethnicity subgroups. Bland–Altman plots were used to analyze further for any trends in errors; for Bland–Altman analyses of differences that were not normally distributed the limits of agreement were based on the 2.5th and 97.5th percentiles of the distribution. Differences between the two algorithm versions were compared using a paired t-test.

To estimate the sample size required for the study, we first conducted an IRB-approved feasibility study with 80 healthy adults. Based on that MAE ± standard deviation (0.96 ± 0.72 breaths/min), a sample size of 50 participants was estimated to provide a power of >0.99 to detect an MAE <3. The power was also >0.99 for both the subgroup of ten healthy participants and the subgroup of 40 with chronic respiratory conditions. If the

MAE and standard deviation were doubled, the power would be >0.99, 0.71, and >0.99, respectively, for the full sample, healthy participants, and those with chronic respiratory conditions.

*User experience survey.* The participants were surveyed about their experience using the app. The questions covered their ease of setting up the phone at the desired angle to capture their face/torso; the clarity of the instructions; their comfort in using the app to assess their general wellness; their comfort in teaching someone else how to use the app; and their expected comfort in using the app several times a day (Supplementary Table 7).

**Reporting summary**. Further information on research design is available in the Nature Research Reporting Summary linked to this article.

## Results

We conducted two separate prospective studies to validate the performance of smartphone-based HR and RR measurements (Fig. 1). The user interfaces of the two custom research apps are shown in Supplementary Fig. 1. The HR algorithm used PPG signals measured from the study participants placing their finger over the rear camera, and the enrollment for the corresponding validation study was stratified to ensure diversity across objectively measured skin tones. The RR algorithm used video captures of the face and upper torso, and the enrollment for the corresponding validation study was stratified to capture participants with and without chronic respiratory conditions.

**Heart rate measurement**. A total of 101 participants were enrolled. Study eligibility criteria are described in Methods (Supplementary Table 1a). After excluding one participant who was found to have an exclusion criterion (pregnancy), there were 100 valid enrollees. Among these, three were withdrawn due to skin tone distribution requirements, and two were withdrawn during data collection due to difficulty in data collection (failure to follow the instruction on holding the phone and a health condition that prevents us from collecting reference HR). Thus, 95 participants completed data collection (Supplementary Fig. 2a). The participants had a mean age of 41.8 years, 75% were female, and skin-tone subgroups were evenly distributed as planned: 33% were subgroup 1 (very light, light, and intermediate), 34% were subgroup 2 (tan and brown), and 34% were subgroup 3 (dark) (Table 1). Skin tone measurement and categorization criteria are described in Methods and Supplementary Table 2.

From these participants, 379 total recordings were attempted. A valid HR was successfully obtained (see details on signal-to-noise ratio [SNR] in Methods) in 361 cases (95.3%). The success rate increased with retries up to three times: 316 measurements (83.4%) were successful on the first try, another 31 measurements (cumulative 91.6%) on the second try, and another 14 measurements (cumulative 95.3%) on the third try. The baseline characteristics of the 14 participants for whom HR values were not successfully reported by the study app for at least one measurement (due to low SNR) did not differ significantly from the remaining participants (Supplementary Table 3). In addition, a corresponding valid reference HR could not be obtained for nine recordings from four participants due to technical error. The remaining 352 recordings with paired valid reference HR contributed to the final analysis (Supplementary Fig. 2a). The average reference HR was 79.8 ± 14.6 beats/min overall, 75.5 ± 11.2 beats/min at rest, and 93.9 ± 15.4 beats/min post-exercise (Supplementary Table 4).

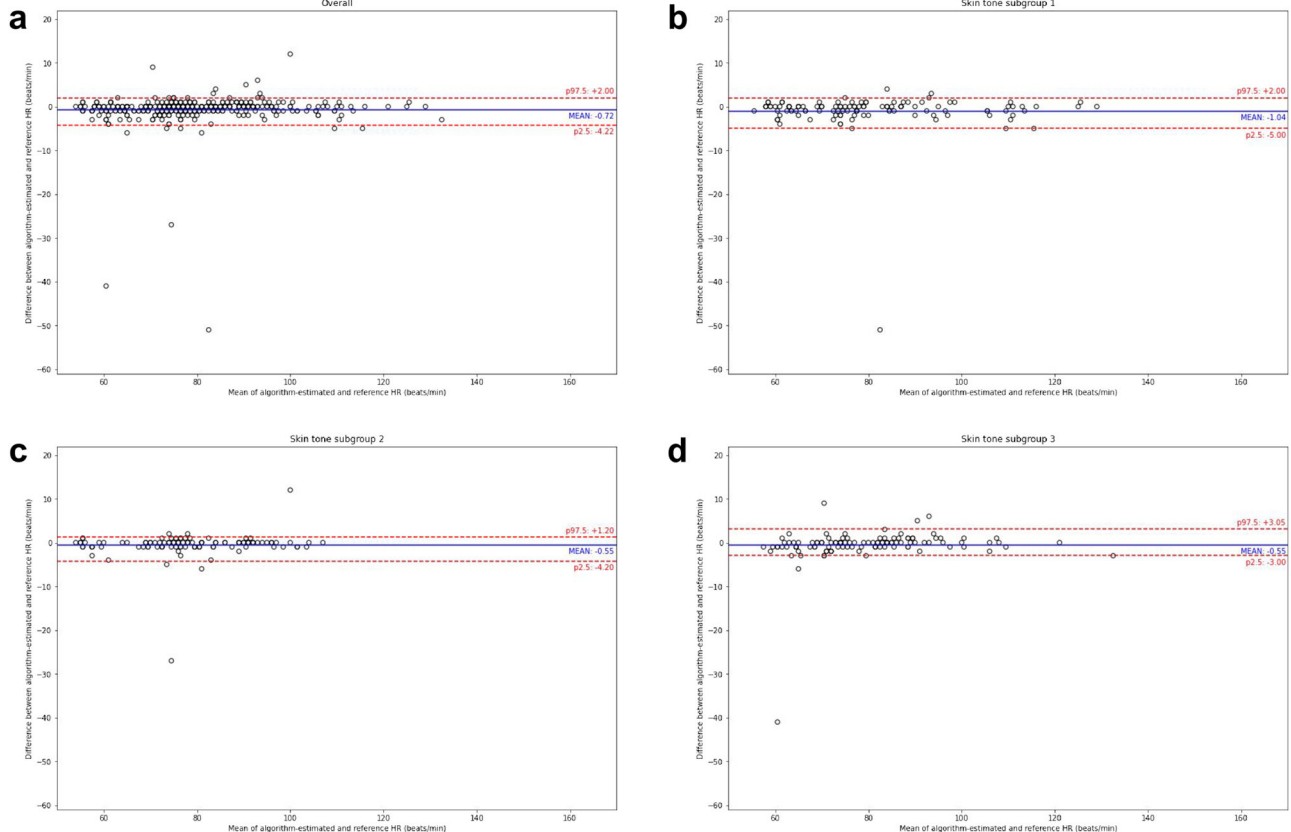

**Fig. 2 Bland–Altman plots for the heart rate (HR) study.** Plots from the full study (**a**) followed by subgroups based on skin type (**b–d**) ($n = 352$ for all [**a**], 119 for subgroup 1 [**b**], 113 for subgroup 2 [**c**], and 120 for subgroup 3[**d**]). The reference HR was obtained from a pulse oximeter (see Methods). Dots represent individual participants; blue lines indicate the mean difference; red lines indicate the limits of agreement (based on the 2.5th and 97.5th percentiles).

Compared to the reference HR, the mean absolute percentage error (MAPE) of the overall study population was 1.63%. The MAPE values for all four data collection windows (including three at rest and one post-exercise measurement) were significantly lower than the prespecified study target of 5% ($p < 0.001$ for all comparisons). The MAPE showed a left-skewed distribution with a long tail (median, 1.14%; range, 0.0–50.6%). The MAPE by skin-tone subgroup was 1.77% for subgroup 1, 1.32% for subgroup 2, and 1.77% for subgroup 3. The MAPE values by skin-tone subgroups were all significantly <5% for each data window (all $P$ values <0.001) (Fig. 1 and Supplementary Table 4). We also found no statistically significant variation in MAPE across the three different lighting conditions.

Figure 2 shows the Bland–Altman plots for comparing the algorithm-estimated HR with the reference HR for the overall population and the three subgroups. Most observations (344/352, 97.8%) were within 5 beats/min. Supplementary Fig. 3 shows the Bland–Altman plots for HR stratified by at-rest versus post-exercise.

**Respiratory rate measurement.** A total of 50 participants were enrolled in the RR study, including ten healthy participants and 40 participants with chronic respiratory conditions, i.e., chronic obstructive pulmonary disease (COPD) or asthma (Supplementary Table 1b and Supplementary Figure 2b). All of the 50 study participants contributed to the final analysis. Self-identified baseline characteristics are presented in Table 1. The mean age was 50 years old; 80% self-identified as White, 14% as Black or African American, and 46% as having Hispanic or Latino ethnicity. The average reference RR was $15.3 \pm 3.7$ breaths/min (Supplementary Table 5).

Both versions of the algorithm successfully estimated RR in all of the study subjects. The mean absolute error (MAE) in the overall study population was $0.84 \pm 0.97$ and $0.78 \pm 0.61$ breaths/min for algorithm versions A and B, respectively (Fig. 1 and Supplementary Table 5), which were significantly lower than the prespecified threshold of 3 breaths/min ($p < 0.001$ for both). Each subgroup also showed MAE values significantly lower than the threshold: algorithm version A, $0.60 \pm 0.52$ breaths/min ($p < 0.001$) for the healthy cohort and $0.90 \pm 1.05$ breaths/min ($p < 0.001$) for the cohort with chronic respiratory conditions; algorithm version B, $0.70 \pm 0.67$ breaths/min ($p < 0.001$) and $0.80 \pm 0.60$ breaths/min ($p < 0.001$), respectively. No statistically significant differences across age and race subgroups were seen (Supplementary Table 6).

Figure 3 shows the Bland–Altman plots for comparing the algorithm-estimated RR with the reference RR for the overall population and the two subgroups. All observations were within 2 breaths/min of the reference RR for algorithm version B, while one observation was >2 breaths/min of the reference RR for version A. The accuracy of the two algorithm versions did not differ significantly ($p = 0.70$).

Supplementary Fig. 4 shows the results from a user experience survey (see Methods) of the study participants. More than 90% of responses were classified as positive, with participants reporting anticipated ease in setting up within a home environment, ease in following the instructions in the app, and comfort using the app to assess general wellness.

## Discussion

We report the results of two prospective clinical studies validating the performance of smartphone algorithms to estimate HR and

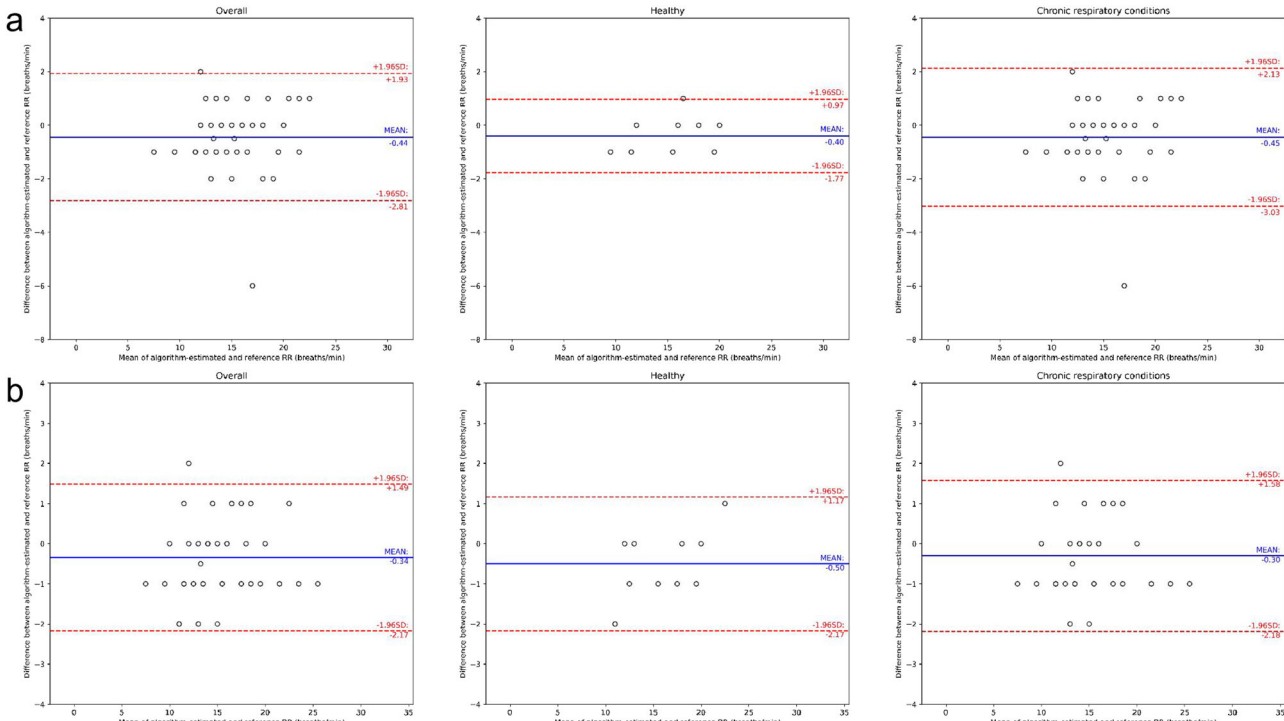

**Fig. 3 Bland–Altman plots for the respiratory rate (RR) study.** Results are presented for **a** algorithm version A ($n = 50$) and **b** algorithm version B ($n = 50$). From left to right, plots are respectively results for the full study followed by subgroups based on absence vs. presence of chronic respiratory conditions. The reference RR was obtained by research coordinators manually counting breaths (see Methods). Dots represent individual participants; blue lines indicate the mean difference; red lines indicate the 95% limits of agreement (mean difference ± 1.96 standard deviations).

RR. Both algorithms showed high accuracy compared to the reference standard vital sign measurements, with the mean MAPE for HR <2% and the mean MAE for RR <1 breaths/min (both significantly below the prespecified targets). In addition, the HR estimation was robust across the full range of skin tones, and the RR estimation generalized to participants with common chronic respiratory conditions: COPD and asthma.

Only a limited number of previous smartphone apps have undergone clinical evaluation for HR measurement[37]. The accuracy of the HR algorithm in this study is especially notable. An MAE less than 5 beats/min or a MAPE less than 10% are standard accuracy thresholds for HR monitors[38,39]. The MAE of 1.32 beats/min in HR is lower than that reported for smartphone apps (2.0 to 8.1 beats/min)[37] and for contemporary wearable devices (4.4 to 10.2 beats/min at rest)[40], albeit with several differences in study design and population. The MAPE of 1.63% is comparable to the performance of current wearable devices. Shcherbina et al. tested six wrist-worn devices and reported a median error <5% for each across various activities and a median error of 2.0% for the best-performing device[41]. Because skin tone can be a potential source of bias in medical devices[22–24], and the accuracy of PPG-based HR estimation can be affected by melanin's light-absorbing property[40,42], we enrolled participants with diverse skin tones to validate the robustness of our HR estimation algorithm across skin tones.

The authors are unaware of any smartphone-based RR measurement apps that have undergone rigorous clinical validation. One previous study tested an algorithm with a similar concept to ours but enrolled only ten healthy subjects[43]. For consumer-grade RR monitoring devices, there is no well-accepted accuracy standard[26]. Our MAE of 0.78 breaths/min is comparable to that of professional healthcare devices, which have reported accuracy of ±2–3 breaths/min[44–47]. One important strength of our approach is determining RR from direct measurement of respiratory motion, rather than trying to derive RR from the variation of PPG-based interbeat interval, which has limitations. The MAE found in this study could be a helpful reference point for future studies. In this study, we tested two algorithm versions for RR estimation that differed only in the SNR threshold. Our results suggest that this parameter had little impact on the accuracy or error rates.

This work supports the use of consumer-grade smartphones for measuring HR and RR. One application of these measurements is in fitness and wellness for the general consumer user. Specifically, an elevated resting HR or slower heart rate recovery after exercise has been linked to lower physical fitness and higher risk of all-cause mortality[48,49]. Evidence suggests the use of direct-to-consumer mobile health technologies may enhance positive lifestyle modification such as increased physical activity, more weight loss, and better diabetes control[50–52]. Tracking one's own health-related parameters over time by the general public can potentially increase motivation for a healthier lifestyle by providing an objective, quantifiable metric[6]. Additionally, there exists strong evidence that regular physical activity is key to improving one's health independent of age, sex, race, ethnicity, or current fitness level for maintaining cardiovascular health[53]. Monitoring one's HR is also an easy and effective way to assess and adjust exercise intensity or enable smartphone-based measurement of cardiorespiratory fitness[54–56].

With further clinical validation across broad populations, such smartphone-based measurement could also be useful in various settings, most notably telehealth where vital sign measurement is challenging due to the remote nature of the patient encounter[57,58]. Though patients can in principle count their own HR or RR, this can be error-prone due to factors such as biases that acute awareness of the self-examination can cause[59,60]. Because the demand for remote triage, diagnosis, and monitoring is burgeoning in the wake of the COVID-19 pandemic, there is increased attention being paid to accurate remote physical examination[3,22,23].

There are several limitations to this work. First, our quantitative results focused on specific study devices (Pixel 3 and 4) and quantitative data on generalization to other devices will be needed. The current studies were also conducted in a controlled setting with structured study protocols. Though the participants used these features without significant study staff assistance, their ease of use in a general population will need further study. For the broad population, there also exists an inherent accessibility/cost and convenience tradeoff between needing to trigger measurements via a camera-equipped smartphone they likely already own, versus receiving passive ongoing measurements via a dedicated wearable wellness tracking device. Next, our reference HR comes from a clinical PPG device instead of an electrocardiogram. There may be infrequent instances of electromechanical dissociation (such as various heart blocks, ventricular tachycardia, etc), in which a pulse rate measured at the periphery may not be the same with the reference electrical "heart" rate[61]. Future evaluations using an electrocardiogram measurement may be helpful in this regard. Our HR algorithm is similarly subject to such errors because it relies on the pulsatile movement of blood in the fingertips. Further, awareness of self-measurement may affect users' RR. A study demonstrated that people may have slightly lower RR when they are aware that they are monitored by observers[62]. It is yet unclear how awareness would affect user-triggered RR measurements using apps in the absence of human-to-human interaction, and creative study designs may be needed to better understand this. In addition, though the study enrollment was optimized for diversity, this impacted the sample size in each subpopulation and the number of covariates that can be analyzed. We did not collect other clinical information such as body mass index, which may potentially affect the accuracy of the algorithms. Larger real-world validation that also measures the effect along such axes will be helpful. Also, these clinical validation studies aimed to evaluate the algorithms at a steady-state; the mean HR post-exercise was 93 beats/min. Further work will be needed to investigate scenarios where HR and/or RR are more elevated and under a broader array of participants, activities, and environments. Though the sex ratio was balanced for the RR study, the female:male ratio was skewed in the HR study. Last but not least, although both HR and RR estimation algorithms met the predefined goals, observations with high deviation from the reference values were still produced by the algorithms (albeit infrequently). Future work to reduce errors is needed.

In addition to the clinical validation studies reported in this work, these HR and RR algorithms are currently undergoing additional broad usability testing across users, different Android devices, and different environments as we make the algorithms more widely accessible for consumers, beginning by incorporating into the Google Fit app.

To conclude, we developed HR- and RR-measurement algorithms for smartphones and conducted two clinical studies to validate their accuracy in various study populations. Both algorithms showed acceptable error ranges with a MAPE under 2% for HR and MAE under 1 breath/min for RR. These algorithms may prove useful in wellness settings such as fitness monitoring. Additional research is warranted before consideration for any future use in clinical settings, such as remote physical examination.

## Data availability

The data are not publicly available because they contain videos of the participants and public release was not part of informed consent.

## Code availability

The algorithms were released as a feature in the Google Fit app to support widespread consumer use as of March 2021. Users and researchers are able to directly access the feature via the app, on supported phones. On the Google Play store, the app can be found at: https://play.google.com/store/apps/details?id=com.google.android.apps.fitness. On the iOS App Store, the app can be found at: https://apps.apple.com/us/app/google-fit-activity-tracker/id1433864494. The raw code is not available because of dependencies on internal proprietary code. The help page for accessing and using the features is at: https://support.google.com/fit/answer/10477667?hl=en. Any future changes made to the algorithm will be indicated in the help notes, update notes, or within the app itself.

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

## Acknowledgements

This work would not have been possible without funding by Google Health and the contribution of many collaborators. We would like to thank Michael Righter and Shilpa Mydur for input on regulatory considerations, Xia Bellavia and Neil Smith for leading the mobile apps testing, Joe Nagle and Hyun Ji Bae for user interface/user experience (UI/UX) designs, Bhavna Daryani for data quality control in the heart rate validation study, Matt Shore for early-stage product development, Rajroshan Sawhney for business development expertize, Benny Ayalew for help in data infrastructure, Justin Tansuwan and Rebecca Davies for visualization tools, Jassi Pannu, Tiffany Kang, and Jacinta Leyden for background research, and Robert Harle and Kapil Parakh for heart rate study protocol feedback. We are also grateful for the valuable manuscript and research feedback from Tiffany Guo, Jacqueline Shreibati, Ronnachai (Tiam) Jaroensri, Jameson Rogers, and Michael Howell. Lastly, we would like to thank Mark Malhotra, Bob MacDonald, Katherine Chou, and (again) Shilpa Mydur and Michael Howell for their support of this project.

## Author contributions

Y.E. and S.Bae developed the HR algorithms; S.Borac, J.Wu., N.W., M.Krainin, and M.R. developed the RR algorithms; J.Wang. and J.Wu developed the mobile apps with the user interface for the clinical studies; J.Wang. developed software infrastructure for algorithms research; S.Bae. and S.Borac. developed software infrastructure for the HR study. E.S.T. provided data collection support for early-stage research; J.A.T., M.J.P., M.Kashyap, and J.C. developed the clinical study protocols; A.M. provided operational support for the clinical validation studies and managed inter-institutional communications; J.A.T. performed statistical analysis for RR study; J.A.T., S.Bae, and E.S.T. performed statistical analysis for the HR study; L.C. implemented the RR study and managed inter-institutional communications; L.C., M.Kashyap, and A.C. implemented the HR study and developed protocol optimizations; M.M. managed contracting and logistics required to begin the clinical validation studies; M.M. and A.C. performed quality control for the HR study data. S.-H.K. and Y.L. provided research feedback and wrote the manuscript with the assistance and feedback of all authors. M.V.M. provided feedback on the HR study protocol and contributed to interpretation of the data and revisions to the manuscript; J.Z. oversaw technical developments and contributed to algorithm development; M.J.P. was the principal investigator for the clinical studies and developed the roadmaps encompassing the studies; S.P. and G.S.C. provided strategic guidance and funding for the project. All authors reviewed and approved the submitted manuscript.

## Competing interests

All authors have completed the ICMJE uniform disclosure form at www.icmje.org/coi_disclosure.pdf and declare: all authors are employees of Google LLC except M.Kashyap and J.C. who are paid consultants of Google via Advanced Clinical; Google LLC may commercialize this technology or have patent rights now or in the future related to this paper. All the authors except M.Kashyap, S.-H.K., and J.C. own Alphabet stock. M.V.M. is a member of the American Heart Assoc Health Tech Advisory Group and Consumer Technology Assoc Health, Fitness, and Wellness standards committees. M.J.P. is a member of the board of El Camino Health, advisory committee of ONC Interoperability Task Force, and scientific advisory board of National Library of Medicine, NIH, the board of AcademyHealth, advisory board of Johns Hopkins University and BME Department, and an advisor, and program committee chair of IEEE EMBS Society.

**Additional information**

