## [Peer Review File · Communications Medicine]

Reviewers' comments:

Reviewer #1 (Remarks to the Author):

This paper validates measuring HRs and RRs using built-in smartphone cameras: HR from a finger over the rear-facing camera and RR from videos sitting in front of the front-facing camera. From 98 participants, HR MAPE was 1.6%, and from 50 participants, MAE was 0.78 breaths/min. This paper also considered a variety of conditions skin tone, flash on/off, light brightness, rest/exercise, and etc. The data collection and evaluation were reasonably performed. Overall, this paper is well written and can be published. However, it also requires major revision.

First of all, I would like to see original or novel methodology in this paper, Regarding most algorithms for HR and RR measurements, I couldn't find any new contributions. Even though this paper focuses on validation, the contributions in terms of the method should be in the paper.

Second, in HR measurement, by considering several ROI regions such as a whole frame, the left half, the right half, the top half, and the bottom half, the authors evaluated SNRs by calculating dominant and non-dominant frequency powers. Then, if none of ROI data had the SNR value equal of higher than 0dB, then the data was discarded. In such a filtering approach, the low MAPE is not surprising.

Third, it's not clear whether the HR and RR were instantaneous values or not. It seems that 30-second data per segment was performed for HR and RR. If so, the results are not that worthy, especially for HR value.

Fourth, comparing the two conditions 1) at rest, flash off, regular light vs. 2) At rest, flash off, dim light, why does condition 1) provides overall worse performances?

Minor

1. (Supplementary Figure 4) The sum of percentages is not 100 for "Easy to teach another person to use the app (47 and 6% only)".
2. Please visualize the RR measuring steps in detail. That part is not easy to follow.

Reviewer #2 (Remarks to the Author):

In this study, investigators evaluate two smartphone camera-based algorithms - photoplethysmography to measure heart rate, and physical motion of the upper torso to assess respiratory rate and thereby assess consumer wellness in a prospective cohort. The study is relevant now in the setting of current pandemic and can be applied to telehealth opportunities and population with scarce resources but have a smart phone with video capabilities. The study is well designed and was able to achieve specific endpoints based on sample size estimations. The app seems simple to follow through and the authors excluded patients with tremors and other conditions who would not be able to participate.

The authors note that the mean absolute percent error was significantly lower than the prespecified error of 5% in the heart rate cohort across different skin tone groups within the 95 participants. MAPE for overall was 1.63% significant for priori of 5%. MAPE has left skewed distribution. MAE of 1.32 beats/min is lower than other reported smartphone apps 2-8.1 and wearable devices 4.4-10.2

beats/min. The app works at different lighting conditions equally well. Similarly, the mean absolute error was lower in the respiratory cohort in both healthy participants and those with chronic respiratory failure. MAE is comparable of 0.78 breaths/min to other devices.

Overall, this is an excellent study with demonstration of newer smart phone-based algorithm. The algorithm as such should be verified by a computational scientist. I would like some clarifications from the authors in terms of analysis.

1. The Bland-Altman analysis shows good representation of agreement/differences. Although there are outliers, as mentioned in the limitation section the analysis would mostly work if the differences are normally distributed. Please provide results of either Shapiro-Wilk test or Kolmogorov-Smirnov test results to ensure the criteria is met.
2. Please provide the R2 and corresponding p value for differences.
3. Similarly, Bland-Altman is not appropriate for repeated measurements. Did the authors add a random effect model to account for the errors?
4. Another major concern is over representation of females in the HR cohort. It should be included in the limitation section.
5. Since this is a clinical validation study, more details about BMI and other basic demographics in each group would be appropriate if available.
6. In the HR cohort the average post exercise HR was 92.9/min. This seems to represent inability to achieve a higher HR with exercise or suboptimal exercise. That should be included as limitation.
7. Ideally HR should be compared to EKG measurement but since the authors chose Masimo MightySat as the reference, please provide details whether the reference was tested against EKG.
8. In the RR cohort, it is uncertain how inherent bias of being watched affected respiratory status. Further please clarify if the app has limitations in terms of detecting respiratory motion in torso with obese patients or those on supplemental oxygen (COPD/Asthma).
9. The hypothesis of the study seeks to demonstrate reliability and consistency. Although the current study provides good accuracy with test results, it will be important to demonstrate reliability with repeated test results in both HR and RR cohorts.
10. Finally, since the app requires operating the video pixels to capture the ROIs in the fingertip and respiratory motion detection, how generalizable is the method to have far impacting results? Authors mention about its application in daily HR, RR measurement for wellbeing assessment and should be mentioned in the limitation as there are other wearable technology which seems to achieve similar results although not with the same error margin but does not require video capture.

Reviewer #1 (Remarks to the Author):

This paper validates measuring HRs and RRs using built-in smartphone cameras: HR from a finger over the rear-facing camera and RR from videos sitting in front of the front-facing camera. From 98 participants, HR MAPE was 1.6%, and from 50 participants, MAE was 0.78 breaths/min. This paper also considered a variety of conditions skin tone, flash on/off, light brightness, rest/exercise, and etc. The data collection and evaluation were reasonably performed. Overall, this paper is well written and can be published. However, it also requires major revision.

[Major comment 1]

First of all, I would like to see original or novel methodology in this paper, Regarding most algorithms for HR and RR measurements, I couldn't find any new contributions. Even though this paper focuses on validation, the contributions in terms of the method should be in the paper.

Response: Thank you for the valuable comment. For the HR estimation algorithm, temporal aggregation/ensemble logic was included. The model estimates a new heart rate with each new frame and caches the value with the associated SNR value. At the end of the session, all the cached values are averaged using cached SNRs as weights. Using integral images enabled running the spatial ensemble model real-time (enable running on lower-end models. In addition, color statistics-based filters were used to detect finger presence. The points have been added as the supplementary methods.

Up to our knowledge, there haven't been RR estimation algorithms that were made public. The algorithm required that the face and upper torso be in the video frame. We have edited the RR algorithm description in a way to better describe its novelties.

Page 12 (RR estimating algorithm)

Before Ensembling was then used to improve the predictive performance. A spectral-spatial ensemble was built in the following way. The respiratory ROI, together with the four quadrants obtained by equally subdividing the ROI defined five regions over which the vertical component of the optical flow was averaged. This resulted in five respiratory waveforms. Next, frequency-domain representations for each of these respiratory waveforms were computed via FFT, from which power spectra were computed. The number of samples in the rolling FFT transform is 900 which provides sufficient resolution for the respiratory rate in the [6, 60] breaths/min range at a video rate of 15 frames per second. The power spectra were then aggregated to obtain a final ensembled power spectrum by adding the five region-level power spectra. Bandpass-filtering was performed to remove low and high frequencies

unlikely to represent valid RRs. Filter cut-off frequencies corresponded to a low of 6 breaths/min and a high of 60 breaths/min. The maximum power frequency and the corresponding SNR value were computed from the ensembled power spectrum. The waveform corresponding to the entire ROI is used for displaying the breathing pattern to the user in the mobile app.

After

Since the vertical component was shown to be sufficient for estimating respiratory rate, only the vertical component was processed in the subsequent steps.

Ensembling was then used to improve the predictive performance. A spectral-spatial ensemble was built in the following way. The respiratory ROI, together with the four quadrants obtained by equally subdividing the ROI defined five regions over which the vertical component of the optical flow was averaged. This resulted in five respiratory waveforms. Next, frequency-domain representations for each of these respiratory waveforms were computed via FFT, from which power spectra were computed. The number of samples in the rolling FFT transform is 900 which provides sufficient resolution for the respiratory rate in the [6, 60] breaths/min range at a video rate of 15 frames per second. The power spectra were then aggregated to obtain a final ensembled power spectrum by adding the five region-level power spectra. The ensembled power spectrum is simply the result of adding the power spectra corresponding to the five regions. Bandpass-filtering was performed to remove low and high frequencies unlikely to represent valid RRs. Filter cut-off frequencies corresponded to a low of 6 breaths/min and a high of 60 breaths/min. The maximum power frequency and the corresponding SNR value were computed from the ensembled power spectrum. The waveform corresponding to the entire ROI is used for displaying the breathing pattern to the user in the mobile app.

[Major comment 2]

Second, in HR measurement, by considering several ROI regions such as a whole frame, the left half, the right half, the top half, and the bottom half, the authors evaluated SNRs by calculating dominant and non-dominant frequency powers. Then, if none of ROI data had the SNR value equal of higher than 0dB, then the data was discarded. In such a filtering approach, the low MAPE is not surprising.

Response: Thank you for the keen comment. Use of SNR cutoff is needed in order to reduce measurement errors. But at the same time, a too strict cutoff may lead to data loss. This study supports that our approach was not very stringent. As described in the manuscript, 101 participants made a total of 379 attempts. Retries up to 3 times were allowed by the study protocol. The first try was successful in 316 measurements (83.4%), the second try in another 31 measurements (cumulative 91.6%), and the final third try in another 14 measurements (cumulative 95.3%). Thus less than 5% of measurements were unsuccessful at the end of this protocol.

In addition, we would like to add additional detail (per comment 1 above). The SNR cutoffs were evaluated after processing each frame; predictions where the SNR exceeded certain thresholds were combined using a weighted average (using the SNR as weights) to estimate the final single HR value for the session. Thus our SNR cutoffs were designed to also make estimations more accurately rather than solely to reject values. Because the SNR cutoff

calculation happened per frame (as opposed to estimating a single SNR value once for the entire session), HR estimation was possible even if there was high SNR in some parts of the session. We have added these details to the supplementary methods for clarity.

[Major comment 3]

Third, it's not clear whether the HR and RR were instantaneous values or not. It seems that 30-second data per segment was performed for HR and RR. If so, the results are not that worthy, especially for HR value.

Response: We appreciate the thorough review. As the reviewer pointed out, the values were not instantaneous. Instantaneous HR values are predicted behind the scenes for each new input frame, but we aggregate instantaneous values throughout the session (weighted average w/ SNR) to estimate a single value for the entire session. This works well for cases when HR is changing such as post-exercise. The SNR evaluated for the entire session once could be low, but SNR evaluated locally can be high - so we can still get high confidence measurements. RR waveform signals were transformed into frequency domain using FFT, with a fallback on time domain.

In clinical settings, HR and RR are usually measured as an average value rather than as an instantaneous value. From a practical viewpoint, this algorithm was built to measure resting HR and thus an average value satisfies the purpose. From a technical perspective, spectral based analyses, which require multiple cycles for estimation, is standard in the field (Tamura et al. Electronics 2014).

[Major comment 4]

Fourth, comparing the two conditions 1) at rest, flash off, regular light vs. 2) At rest, flash off, dim light, why does condition 1) provides overall worse performances?

Response: This is a very nuanced point. Indeed, there is a trend toward a lower MAPE at data window 1b (rest, flash off, regular light) than 1c (at rest, flash off, dim light). It was mainly driven by a single outlier in data window 1b (at rest, flash off, led normal).

Excluding that single outlier brings the MAPE down to a similar level.

1b MAPE: 1.64%

1b MAPE (excluding 1 outlier): 1.32%

1c MAPE: 1.33%

[Minor comment 1]

(Supplementary Figure 4) The sum of percentages is not 100 for "Easy to teach another person to use the app (47 and 6% only)".

Response: Thank you for the detailed review. This was a typo where instead of 47 cases (out of 50, or 94%), we had input 47% for the plot. This has now been corrected.

[Minor comment 2]

Please visualize the RR measuring steps in detail. That part is not easy to follow.

Response: RR estimating algorithm has been revised as described in major comment 1. RR data collection has been modified as shown below.

Page 12 (RR data collection)

Before The in-app instructions guided the participants to wait several minutes after any active movement and to stay comfortable and breathe normally during the measurements.

After The in-app instructions guided the participants to wait several minutes after any active movement. The participants were encouraged to stay comfortable and breathe normally during 30 seconds of the measurements.

Reviewer #2 (Remarks to the Author):

Overall, this is an excellent study with demonstration of newer smart phone-based algorithm. The algorithm as such should be verified by a computational scientist. I would like some clarifications from the authors in terms of analysis.

[Comment 1]

The Bland-Altman analysis shows good representation of agreement/differences. Although there are outliers, as mentioned in the limitation section the analysis would mostly work if the differences are normally distributed. Please provide results of either Shapiro-Wilk test or Kolmogorov-Smirnov test results to ensure the criteria is met.

Response: We thank the reviewer for identifying this issue. The P value for the Shapiro-Wilk test of the differences in the Bland-Altman analyses for all of the HR groups in Figure 2 and for those with algorithm A for RR are $< .05$. Because of the this non-normal distribution, in the revised manuscript, these Bland-Altman plots have been updated to have the 2.5%ile and 97.5%ile values used to define the limits of agreement, a non-parametric approach suggested by Bland and Altman (Statistical Methods in Medical Research, 1999;8:135-160).

For the RR algorithm, the P values for the Shapiro-Wilk tests for the distribution of differences met the criteria for normality (P values= 0.29, 0.58 and > 0.99 , respectively for all, healthy and “sick” participants). These Bland Altman analyses were not updated.

For the Bland-Altman analyses in Supplemental Figure 3 (HR at rest and post-exercise), the P values for the Shapiro-Wilk test of the differences were all $< .05$, except for at rest, skin group 2 (P=.06). The limits of agreement for all the Bland-Altman plots are now the 2.5%ile and 97.5%ile, except for HR at rest, skin group 2 are unchanged.

[Comment 2]

Please provide the R^2 and corresponding p value for differences.

Response: The R^2 and P values for the differences for the RR Bland Altman analyses are shown in the table below.

Group	R2 (P value)
All - algorithm A	.009 (P = .51)
Healthy	.032 (P=.62)
Chronic respiratory conditions	.072 (P = .60)
All - algorithm B	.013 (P = .44)

Healthy- algorithm B	0.54 (P = .11)
Sick-algorithm B	.001 (P = .84)

[Comment 3]

Similarly, Bland-Altman is not appropriate for repeated measurements. Did the authors add a random effect model to account for the errors?

Response: The mean difference for the Bland-Altman analyses that included multiple observations per participant have been changed to reflect the results from random effects model (with participant as the random effects term).

For the HR data, a mixed effects model was used, with difference and average heart rate (algorithm-derived HR + ground truth HR)/2 as fixed effects terms and participant as a random effects term. This method does not readily lend itself to computation of R², but the P values for average heart rate as a predictor of difference were: all, P = .18; skin group 1. P = .88; skin group 2, P = .12; and skin group 3- P = .04.

For Bland-Altman analyses of HR post-exercise, with one observation/participant, the R² and P value are shown in the table below.

group	R ² (P value)
Post-exercise all	.027 (P = .13)
Post-exercise, skin group 1	.031 (P = .35)
Post-exercise, skin group 2	.019 (P = .49)
Post-exercise, skin group 3	.097 (P =.1)

For the analyses of HR data at rest, with multiple values/participant, th P values for average HR as a predictor of difference were: all, P= .06; skin group 1, P =.04; skin group 2, P = .71; skin group 3, P = .025.

We are leaving these out for brevity and clarity, and defer to the editor on any preferences for inclusion.

As we state in the manuscript, the results were virtually the same because of the limited effect from clustering within participants.

[Comment 4]

Another major concern is over representation of females in the HR cohort. It should be included in the limitation section.

Response: The point has been added in the limitations section as shown below.

Page 7 (Limitations)

Though the sex ratio was balanced for the RR study, the female:male ratio was skewed in the HR study.

[Comment 5]

Since this is a clinical validation study, more details about BMI and other basic demographics in each group would be appropriate if available.

Response: Thank you for the comment. Unfortunately, information other than described in Table 1 was not collected. In retrospect, we also assume that subgroup analysis of the RR data according to BMI would be interesting. However, this was a prospective study and we do not have any means to collect additional data. This issue has been added to the limitations section.

Page 7 (Limitations)

Before Larger real-world validation that also controls for additional factors such as body-mass index will be helpful.

After We did not collect other clinical information such as body mass index, which may potentially affect the accuracy of the algorithms. Larger real-world validation that also measures the effect along such axes will be helpful.

[Comment 6]

In the HR cohort the average post exercise HR was 92.9/min. This seems to represent inability to achieve a higher HR with exercise or suboptimal exercise. That should be included as limitation.

Response: The point has been added in the limitations section as shown below.

Page 7 (Limitations)

Before Also, these clinical validation studies aimed to evaluate the algorithms at a “steady state” and further work will be needed to investigate acute clinical scenarios such as elevated or depressed HR or RR in urgent medical situations.

After Also, these clinical validation studies aimed to evaluate the algorithms at a “steady state”; the mean HR post-exercise was 93 beats/min. Further work will be needed to investigate scenarios where HR and/or RR are more elevated, and under a broader array of participants, activities, and environments.

[Comment 7]

Ideally HR should be compared to EKG measurement but since the authors chose Masimo MightySat as the reference, please provide details whether the reference was tested against EKG.

Response: This is an important point, that we have revised the limitations to help emphasize.

Discussion (page 7)

Before Next, our reference HR comes from a clinical PPG device instead of an electrocardiogram. There may be infrequent instances of electromechanical dissociation (such as various heart blocks, ventricular tachycardia, etc), in which a pulse rate measured at the periphery may not be the same with the reference electrical “heart” rate.⁴³ Our HR algorithm is similarly subject to such errors because it relies on the pulsatile movement of blood in the fingertips.

After Next, our reference HR comes from a clinical PPG device instead of an electrocardiogram. There may be infrequent instances of electromechanical dissociation (such as various heart blocks, ventricular tachycardia, etc), in which a pulse rate measured at the periphery may not be the same with the reference electrical “heart” rate.⁴³ Our HR algorithm is similarly subject to such errors because it relies on the pulsatile movement of blood in the fingertips. Future evaluations using an electrocardiogram measurement may be helpful in this regard.

[Comment 8]

In the RR cohort, it is uncertain how inherent bias of being watched affected respiratory status. Further please clarify if the app has limitations in terms of detecting respiratory motion in torso with obese patients or those on supplemental oxygen (COPD/Asthma).

Response: We appreciate the reviewer’s comment. Potential impact of being watched has been revised to be clearer in the limitations section.

Discussion (page 7)

Before	Further, awareness of self-measurement may affect users' RR. A study demonstrated that people may have lower RR when they are aware that they are monitored by observers. ⁴⁴ It is yet unclear how awareness would affect RR measurements using apps in the absence of human-to-human interaction.
After	Further, awareness of self-measurement may affect users' RR. A study demonstrated that people may have slightly lower RR when they are aware that they are monitored by observers. ⁴⁴ It is yet unclear how awareness would affect user-triggered RR measurements using apps in the absence of human-to-human interaction, and creative study designs may be needed to better understand this.

Unfortunately, BMI/obesity data were not collected and patents on supplemental oxygen were excluded by the eligibility criteria. It is still unclear how obesity would affect the RR measurement accuracy (see our response above to [Comment 5]). But we anticipate the need for supplemental oxygen would have little impact.

[Comment 9]

The hypothesis of the study seeks to demonstrate reliability and consistency. Although the current study provides good accuracy with test results, it will be important to demonstrate reliability with repeated test results in both HR and RR cohorts.

Plan: Thank you for the insightful comment. We agree this is useful - hence multiple conditions were tested for both the HR and RR algorithms, which measures the combination of varying conditions and repeated attempts by the same subjects. As described in Supplementary Table 4, participants did four measurements for the HR algorithm. Also for the RR algorithm, we tested two different versions and confirmed the reliability. In addition, we tried to enroll participants from various populations in both studies.

That said, we agree that reliability and consistency could be confirmed by future studies with a broader array of activities, participants, and environments. This point has been added in the limitations section (please see comment 6 above).

[Comment 10]

Finally, since the app requires operating the video pixels to capture the ROIs in the fingertip and respiratory motion detection, how generalizable is the method to have far impacting results? Authors mention about its application in daily HR, RR measurement for wellbeing assessment and should be mentioned in the limitation as there are other wearable technology which seems to achieve similar results although not with the same error margin but does not require video capture.

Plan: This is an important tradeoff for consumers: whether to use a smartphone (and its sensors) that users likely already have, versus buying a dedicated wearable for tracking (that they would have to charge separately but that may potentially have a different set of measurement capabilities or accuracies).

Discussion (page 7)

For the broad population, there also exists an inherent accessibility/cost and convenience tradeoff between needing to trigger measurements via a camera-equipped smartphone they likely already own, versus receiving passive ongoing measurements via a dedicated wearable wellness tracking device.

REVIEWERS' COMMENTS:

Reviewer #1 (Remarks to the Author):

The revised manuscript addressed the issues appropriately. I would like to thank all authors for their efforts. It is now suitable for publication.

Reviewer #2 (Remarks to the Author):

The authors have addressed all my concerns. I do not have any further comments.